# Matrix Metalloproteinases and Heart Transplantation—A Pathophysiological and Clinical View

**DOI:** 10.3390/medicina59071295

**Published:** 2023-07-13

**Authors:** Gabriela Patrichi, Andrei Patrichi, Catalin-Bogdan Satala, Anca Ileana Sin

**Affiliations:** 1Department of Cell and Molecular Biology, George Emil Palade University of Medicine, Pharmacy, Science and Technology of Targu Mures, 540142 Targu Mures, Romania; anka_sinn@yahoo.com; 2Department of Pathology, Clinical County Emergency Hospital, 540136 Targu Mures, Romania; patrichi_andrei@yahoo.ro (A.P.); stlcatalin92@yahoo.com (C.-B.S.); 3Department of Pathology, George Emil Palade University of Medicine, Pharmacy, Science and Technology of Targu Mures, 540142 Targu Mures, Romania

**Keywords:** matrix metalloproteinases, heart transplantation, graft rejection

## Abstract

Heart transplantation is undergoing a continuous development, with rates of success increasing substantially due to advances in immunosuppressive therapy and surgical techniques. The most worrying complication occurring after cardiac transplantation is graft rejection, a phenomenon that is much affected by matrix metalloproteinases (MMPs), with the role of these proteases in the cardiac remodeling process being well established in the literature. A detailed investigation of the association between MMPs and cardiac rejection is necessary for the future development of more targeted therapies in transplanted patients, and to discover prognostic serum and immunohistochemical markers that will lead to more organized therapeutic management in these patients. The aim of this review is therefore to highlight the main MMPs relevant to cardiovascular pathology, with particular emphasis on those involved in complications related to heart transplantation, including cardiac graft rejection.

## 1. Introduction

Heart transplantation techniques are undergoing continuous development and innovation, including new targeted drugs and therapeutic strategies that confer improved prognosis and longer life expectancy to patients compared to previous decades [1].

This progress in heart transplantation is due primarily to a better understanding of the mechanism of graft rejection, and a significant contributor to this phenomenon are matrix metalloproteinases (MMPs).

A detailed study of these proteases is required, as they will contribute to the establishment of targeted treatments for better therapeutic outcomes in the future [2].

The main complications following heart transplantation are graft rejection, infectious complications, conduction disorders, vasculitis, and coronary heart graft disease (including vessel remodeling and obstruction by intimal hypertrophy, as well as systemic complications including hypertension, renal failure, metabolic disorders, and hyperuricemia due to immunosuppressive treatment, and also induced cancers).

Patient quality of life and survival time have improved remarkably due to the strict selection criteria of donors and recipients, through improved treatment guidelines, and through rigorous monitoring of patients (the latter of which occurs throughout their lives, but especially in the first six months following the procedure). Therefore, special attention is focused on rejection, which can occur at any time after heart transplantation.

It is essential to correctly diagnose the degree of rejection, both for further management of the lesions and for a better understanding of the attendant biological and molecular changes [3]. Two types of rejection are known: cell-mediated rejection and antibody-mediated rejection (CMR and AMR, respectively). In the acute phases, from a histopathological point of view, CMR is characterized by the appearance of abundant mononuclear inflammatory infiltrate, myocyte damage, and fibrotic changes, and these processes may continue with intimal thickening of the coronary arteries in chronic stages [4,5]. Staging of this rejection phenomenon is based on serial endomyocardial biopsies collected from the patient, using the grading system established by the International Society for Heart and Lung Transplantation (ISHLT).

Activation of T lymphocytes plays a central role in initiating an immune response that can trigger a rejection reaction. Following ischemia and reperfusion injury, sometimes expressed, this induces myocyte necrosis lesions and myocyte degenerative changes, such as myocyte premiocytosis, cytoplasmic microvacuolysis, and fine granular cytoplasm, with the structure of the cardiac muscle being alarmingly altered; this leads to wide-area alterations in myocardial structure, which also implies hemodynamic perturbations. Cardiac remodeling inevitably occurs, and MMPs play an important role in the dynamics of this process; in recent years, real progress has been made in the study of these proteases and their role in cardiac remodeling [6]. Changes in the dynamic components of the extracellular matrix potentiate the inflammatory response and the reparative pathways of myocardial tissue, and its role in the process of cell turn-over and in maintaining cardiac structure and integrity is well known [7].

Cardiac remodeling involves alterations in the size, shape, and function of heart tissue, as generally induced by a cardiac lesion. These lesions affect molecular, cellular, and interstitial cardiac structure, as well as gene expression. It occurs in many cardiovascular diseases, but especially in graft rejection, and MMPS are essential to the pathophysiological mechanisms of this remodeling [8].

Cardiac graft rejection is mediated by the appearance of an inflammatory T-cell infiltrate at the level of the graft, immediately followed by the initiation of the immune response and destruction of the graft. This is accompanied by alteration of extracellular matrix components, with this turnover of matrix proteins being regulated by MMPs [9].

## 2. Aim

The purpose of this review is to present the main MMPs involved in cardiovascular pathology, and to highlight the role of different MMPs in post-transplant complications using the existing data from the literature.

## 3. MMPs: A General Introduction

MMPs are a family of 25 proteolytic enzymes (zinc-dependent endopeptidases) present in the extracellular matrix whose known role is to degrade its structural components [10]. A primary role of these proteases has been that of cleaving extracellular matrix proteins under certain physiological and pathological conditions [11]. Physiologically, they are involved in different processes, mediating different activities of cell proliferation and differentiation and tissue repair, as well as in the mechanisms of apoptosis and angiogenesis, or cell migration [12].

MMP activity is tightly regulated by proteolytic activation and is inhibited by tissue inhibitors of metalloproteinases (TIMPs), and an imbalance between these two components results in the development of many diseases, including cardiovascular diseases, neurodegenerative diseases and cancer, but also in inflammatory processes such as arthritis or fibrotic changes. Both intracellular and extracellular MMPs are responsible for the development of cardiovascular disease—including post cardiac transplant rejection [10,13].

In addition to TIMPs, the expression and the activity of these enzymes can also be influenced by other metabolic pathways, chemical agents and cell signaling molecules (different hormones, growth factors, and cytokines such as insulin or insulin-like growth factor-1, leptin, adiponectin, and corticosteroids). Moreover, gene expression was also found to be involved in the upregulation of MMPs [14].

## 4. MMP Classification

MMPs are part of the methzicin family, a superfamily of zinc-dependent endopeptidases present in the extracellular matrix, along with a-disintegrins and metalloproteinases (ADAMs) and a-disintegrin and metalloproteinase with thrombospondin motifs (ADAMTS) [15].

These proteases have been classified in terms of substrate affinity, and based on the mechanism of enzymatic reactions and the transmembrane or soluble domain, as well as their structure or localization [10].

In terms of structure, MMPs share a common, three-dimensional pattern consisting of a propeptide sequence, a catalytic domain and a hemopexin-like c-terminal domain attached to the catalytic domain by a flexible hinge region [16].

Based on affinity for the structural domain of extracellular matrix components and subcellular distribution, MMPs have been divided into membrane-bound MMP-MT-MMPs, and collagenases, gelatinases, stromelysins and matrilysins, respectively. Collagenases include MMP1, MMP8, MMP13, and MMP18, and play a fundamental role in bone and ligament pathophysiology. The gelatinases MMP2 and MMP9 are known to play a role in angiogenesis and neurogenesis and through their various biological functions there are studies that demonstrate their involvement in the atherosclerotic process or in the metabolic syndrome, along with the collagenases MMP1 and MMP8 mentioned above [17]. The stromelysins MMP3, MMP10, and MMP11 have a fundamental role in extracellular matrix degradation, while the matrilysins MMP7 and MMP26 are known to play a role in the fibrosis process. Of these, we focus special attention on gelatinases, especially MMP2 and MMP9, as their contribution to cardiovascular pathology is also well known [18].

In addition to the known roles of these proteases, we pay special attention to their potential use as biomarkers, essential tools in predicting and detecting changes in myocardial structure that lead to cardiac and vascular pathologies [19]. These biomarkers, whether serum or immunohistochemical, may be potential therapeutic targets in end-stage heart failure.

Numerous MMPs present in cardiomyocytes have been reported, among them interstitial collagenases (MMP1, MMP13), stromelysins (MMP3), and gelatinases (MMP2, MMP9) whose function and structure is extensively presented in this article [20]. These proteases have also been implicated in cardiovascular pathology, of which the most responsible MMPs involved in cardiac allograft rejection pathology are MMP2 and MMP9.

Therefore, we summarize MMPs with relevance in cardiovascular pathology:

MMP1 is part of the collagenase group and is mainly expressed in fibroblasts, but also in leukocytes or endothelial cells. In the myocardium, MMP1 expression is known to be increased, especially at 4 days post myocardial infarction. It preferentially degrades collagen types I and III and has an affinity for fibrillar collagen [21].

MMP2 is a gelatinase, which together with MMP9 constitutes the main proteases involved in cardiac rejection pathology. It is known to disintegrate extracellular matrix proteins in human myocardium. MMP2 expression has been found in cardiomyocytes, endothelial cells, vascular smooth muscle cells, macrophages, and fibroblasts. Pathologically, increased plasma expression of MMP2 has been demonstrated post myocardial infarction and at the infarct site due to stimulation of cardiac myocytes and fibroblasts. Finally, increased levels of MMP2 were observed in cardiac damage in both human subjects and animals [22,23].

MMP3 is part of the stromelysin family and is secreted by cardiac fibroblasts and macrophages. It has multiple roles, including the degradation of extracellular matrix components—collagen, fibronectin, laminin, proteoglycans, and vitronectin. Elevated levels of MMP3 correlate with the severity of myocardial infarction and are a prognostic and prevalence index of cardiac damage [24,25,26].

MMP7 is from the matrilizin group and cleaves certain extracellular matrix proteins, as well as activating MMP1, -2, -8, and -9. It is considered a strong TIMP inhibition resistor when compared to the other MMP types. It is expressed by cardiomyocytes, endothelial cells, and macrophages [27]. It plays an important role in the left ventricular remodeling process and has been shown to be responsible for the occurrence of post myocardial infarction arrhythmias, decreasing post myocardial infarction survival rate and thus increasing hospitalization rate due to cardiac damage [28,29].

MMP8 is also a collagenase secreted mainly by neutrophils and macrophages with important implications in inflammatory response, ventricular remodeling, cardiac rupture, and end-stage heart failure.

MMP9, together with MMP2, is a highly important gelatinase secreted by cardiomyocytes, endothelial cells, neutrophils, macrophages, and fibroblasts. It is considered an independent predictor of prognosis and mortality from cardiovascular disease, and increased levels of MMP9 are associated with ventricular dysfunction [30,31].

The main proteases involved in cardiovascular pathology will be listed in Table 1, respectively their function, location and structure.

## 5. MMP Role and Localization

Initially discovered extracellularly, as matrixin proteases interacting with components of the extracellular matrix, over time, these enzymes were demonstrated to have ubiquitous roles and a wide variety of functions.

Two main types of MMPs were initially discovered—those located in the extracellular environment and released in a latent proenzymatic form (proMMP), and membrane bound MT-MMPs [18]. It has been demonstrated in various studies that MMP plays an important role in physiological remodeling processes, tissue morphogenesis, and tissue repair. An increase in MMP expression has also been found in tumor pathogenesis, especially in tumor angiogenesis and metastasis processes, in hyperplastic intimal proliferation processes, and in plaque rupture in atherosclerosis and rheumatoid arthritis. An increase in MMP activity has also been reported in cardiac morphogenesis and cardiovascular diseases [20,21].

More recent studies have revealed the existence of these proteases not only in the extracellular matrix, but also in the intracellular environment, the latter having a huge role in the pathogenesis of various diseases, especially cardiovascular, renal, and inflammatory diseases, as well as tumors. Intracellular MMP can exert their role in at least two different mechanisms: protease-dependent and protease-independent mechanisms [11].

Biological research on the activity of these MMPs has revealed their multiple roles in the diagnosis of various diseases and in the understanding of certain pathomechanisms, being useful tools in the design of targeted therapies [6,32].

MMP2 and MMP9 belong to the class of gelatinases that share a substrate (collagen IV) and are structurally homologous [33].

In general, MMP activity is regulated by transcriptional processes and the activation of zymogenic precursors, as well as their action on and interaction with extracellular matrix components. Control of the activity of these proteases is also achieved by the aforementioned TIMPs, which represent an endogenous class of low-molecular weight molecules [34,35].

Under normal conditions there is a balance between MMP and TIMP activities; the occurrence of any change in the composition of the extracellular matrix (such as activation of the inflammatory process or triggering of tissue remodeling phenomena) causes an increase in MMP expression and thus an imbalance in the MMP/TIMP ratio [10,36].

Therefore, an important role in maintaining cardiac function is played by the interaction between MMPs and TIMPs through the process of ventricular remodeling. The inhibitory action of TIMPs are predominantly at the level of endothelial cells, and the use of these markers for therapeutic purposes show great potential [37].

The role of MMPs and TIMPs in maintaining the balance between disease onset and health status is well known, and a good understanding of the mechanisms of action of these proteases is desirable. Thus far, certain regulatory abnormalities between MMPs and TIMPs have been show to lead to the development of numerous pathological conditions, including inflammatory processes, tissue destruction, alteration of immune mechanisms and extracellular matrix components, fibrosis processes, angiogenesis, and carcinogenesis [19,38].

## 6. MMPs and Cardiovascular Disease

Cardiovascular disease is a major concern and remains the leading cause of morbidity and mortality worldwide. In addition to the numerous known risk factors that favor the development of various heart diseases, MMPs also contribute to the development of these pathologies. Thus, alterations in the expression of these proteases lead to increased risk of cardiovascular morbidity and mortality.

MMP has been shown to present broad roles in the pathogenesis of multiple cardiovascular diseases, not limited to transplant pathology. Among these, we mention an important contribution in the process of atherosclerosis or in myocardial ischemia and reperfusion lesions. Multiple studies also show the involvement of MMPs in diabetes, myocardial infarction, aneurysm, hypertension, and cardiomyopathies [39,40,41].

### 6.1. Atherosclerosis

MMPs have been associated with atherosclerosis, a degenerative disease characterized by lipid accumulation on the intimal surface of the arteries, with the formation over time of atheromatous plaques by various promoting factors. These proteases have the ability to degrade atheromatous plaques, thereby destroying the structure of the vascular wall [42].

Moreover, Lee et al. demonstrated (in a study on atherosclerosis) the involvement of MMPs in inflammatory processes, and showed that the decrease in the degree of inflammatory infiltrate is due to inhibition of the elastase system through activation of MMPs, triggering the production of local growth factors and creating a vicious circle of initiating a new inflammatory process with a role in atherosclerotic plaque formation [43].

MMP2, MMP3, MMP9, and MMP14 are thought to promote vascular damage by their proaterogenic and proinflammatory effects. They act at the level of both smooth muscle cells and endothelial cells [44]. Increased levels of MMP2 and MMP9 degrade the extracellular matrix and lead to rupture of vulnerable arterial plaques, increasing plaque instability [14]. Thus, these protease actions together contribute to complications secondary to atherosclerotic lesions.

There are also numerous trigger factors for plaque destruction, including influenza virus (which is associated with overexpression of MMP13), and carotid stenosis (associated with increased levels of MMP1, MMP7, and MMP10) [11].

Numerous studies in rats have revealed that MMP2 overexpression is frequently found in various cardiac pathomechanisms resulting in troponin I proteolysis, cardiac mitochondrial dysfunction, left ventricular remodeling, and systolic heart damage [45,46].

MMP2 is known to be involved in the degradation of cardiac sarcomeric proteins such as troponin I, desmin, and alpha actin, leading to contractile dysfunction, as well as in vascular smooth muscle cells, where it produces angiotensin II-mediated damage with secondary appearance of inflammation and triggering of the immune response [47].

### 6.2. Ischemia and Reperfusion Injuries

MMP2 is activated in the myocardium following cytotoxic anthracycline treatment in ischemia and reperfusion injury, and can also be triggered by certain proinflammatory cytokines or by diabetic cardiomyopathy, through degradation of cardiac proteins or impairment of myocardial contractility [48,49].

Disruption of the major compatibility system with loss of myosin filaments and sarcomere disarrangement underlies the pathomechanism of dilated cardiomyopathy, with MMP9 also contributing [50].

In myocardial ischemia and reperfusion injury, MMP2 acts at the intracellular level by degrading proteins responsible for myocardial contractility, causing a decrease in the affinity of contractile myofilaments for calcium, with the secondary appearance of contractile dysfunction [51].

### 6.3. Hypertension

Hypertension is a common condition in the population, characterized by increased systolic and diastolic pressure above 140/90 mmHg. The presence of this disease underlies the development of other known cardiovascular diseases. An alteration of the large vascular walls, consequent with microcirculatory damage and the appearance of inflammation, is determined by the action of MMP2 and MMP9 on endothelial cells with stimulation of the release of pro-inflammatory factors [41].

Elevated levels of MMP and TIMP1 activity have been observed in patients with primary hypertension associated with vascular wall thickening and stiffening [52].

MMP9 acts at the level of the extracellular matrix in arterial hypertension, through collagen degradation consequent to vascular wall alteration and the development of hypertension. It is also involved in the process of arterial wall remodeling in myocardial vessels as a result of increased blood pressure and altered wall elasticity. Implicitly, this compensatory remodeling affects both the structure of the myocardium and the vessels, creating a vicious circle that in time will lead to other cardiovascular pathologies [53].

Various studies have reported increased levels of MMP9 and TIMP1 in hypertensive patients without secondary cardiac damage, and increased levels of MMP2 in patients with cardiac damage.

MMP1 and TIMP1 activity correlates with the extent of organ damage in hypertensive patients [19].

Other studies have revealed that increased levels of MMP7, MMP9, and TIMP1 are predictive factors in the development of left ventricular hypertrophy [54].

If these proteases play an important role in the pathogenesis of cardiovascular disease, the use of MMP inhibitors could be therapeutically beneficial in preventing the development of complications.

### 6.4. Myocardial Infarction

Increased levels of MMP9 have also been reported in induced myocardial infarction in mice [55].

MMPS play an important role both in atherosclerotic plaque stability (when maintained within normal limits) and in vessel rupture and obstruction with secondary ischemia and necrosis (when there is increased expression of the activity of these proteases). Therefore, even in myocardial infarction, it is necessary to have a balance between MMPs and TIMPs to limit the occurrence of this pathology. However, when this occurs, an inflammatory infiltrate accumulates at the site of myocardial necrosis, predominantly with neutrophils but also with inflammatory mediators that determine the activation of MMP9 which in turn contributes to the pathomechanism of post-infarction ventricular remodeling and secondary to terminal cardiac damage [56]. As MMP9 is secreted by neutrophils, it follows that high levels of this protease is present in myocardial infarction and participates in the process of post-infarct remodeling, impeding tissue repair and angiogenesis [57].

MMPs make an important contribution to the ventricular remodeling process after myocardial infarction or viral infection by contributing to the cleavage of collagen and elastin structure, ultimately leading to hypertrophy and compensatory dilation [6,58]. These changes result in end-stage heart failure and, ultimately, death [4].

### 6.5. Aortic Aneurysm

Aortic aneurysm is a dilatation of the vascular wall due to alteration of structural components of the wall (collagen and elastin). It is a condition not to be due to its significant complications, with the risk of wall emboli and even rupture. Thus, MMP2, MMP9, and MMP12 have been shown to contribute to the degradation of the extracellular matrix by decreasing elastin levels and increasing collagen levels. Consequently, an inflammatory process occurs, with accumulation of macrophages, neutrophils, LT and pro-inflammatory cytokines leading to fragmentation of elastic fibers, fibroblastic proliferation or fibrosis, and, in more severe stages, wall destruction [59].

### 6.6. Cardiac Rejection Pathology

In cardiac rejection pathology, the influx and persistence of raised levels of CD3 and CD68 inflammatory infiltrate aggravates the acute cellular rejection stage of the graft and, implicitly, the increased degree of rejection is associated with high levels of pro-inflammatory cytokines (IL-6, TNF alpha, TGF beta) and an elevated level of MMP9; however, not of TIMP1, which has an anti-inflammatory role. Thus, MMP9 is responsible for the influx of LT and monocytes into the transplanted heart and is considered a marker for the inflammatory response in cardiac rejection. MMP9 activity has been found to increase in direct proportion to the degree of rejection, with peak levels of MMP9 expression occurring in advanced stages of rejection, and it may in the future be considered as an important parameter to be included in standard evaluation and diagnostic criteria [5,60,61].

Another important parameter to follow in the rejection phenomenon is the process of fibrosis. Elevated expression of MMP2 and MMP9, as well as TIMP1 and TIMP2, correlates with increased expression of collagen types II and III, which in turn progressively increase with the degree of graft rejection. TIMP1 and TIMP2 activate fibroblast proliferation in the extracellular matrix and regulate collagen synthesis and production in the transplanted heart graft. MMP9 is considered a surrogate marker for the degree of inflammation in the transplanted heart, and is used to distinguish between a rejected and an unrejected heart [5,62,63].

As emphasized in this review, several MMPs have been associated with cardiovascular disease, and are outlined in Figure 1.

## 7. Discussion

Past and current research has revealed the potential implications of these matrix metalloproteinases in cardiovascular pathology, including transplantation. Understanding the role of these enzymes during these pathological processes remains an important issue. A continuity of research is necessary considering the role of these proteases in the multiple pathologies mentioned in this review; therefore, it is very important to review all factors influencing MMP activity, the main MMPs related to cardiovascular diseases and to focus on what is known today regarding heart transplantation.

Our goal was to highlight the specific roles for these MMPs in the pathogenesis of cardiovascular diseases, especially cardiac allograft rejection. Thus, MMP2 and MMP9 have been demonstrated to play an important role in heart transplant pathology and its complications.

Vanhoutte et al. (2013) published a cohort study on a population of 39 transplanted patients in the first postoperative year in which they demonstrated, on serial biopsies, that the expression of these proteases, in particular MMP2 and MMP9 but also of their TIMP1 inhibitors, correlated with the degree of graft rejection. Even though some studies postulated that peak levels of MMP expression were observed in patients at the most advanced stage of acute cellular and humoral rejection, i.e., ISHLT (International Society for Heart and Lung Transplantation) stage III [5], the results are contradictory, because there are still many limitations regarding their use as clinical biomarkers. Up to date, it has been demonstrated that the prognostic utility of MMP serum levels has a greater impact when correlated with other biomarkers, such as VEGF-A (Vascular Endothelial Growth Factor A) [64]. This topic raises concerns and demonstrates the need for further studies targeting the activity of these proteinases in both early and late post-transplant patients.

Various studies in mice have also shown that increased expression of MMP2 and MMP9 have a negative influence on graft survival, while tissue inhibitors of these proteases, in particular TIMP1, increase survival due to suppression of the inflammatory response [60,61].

Moreover, studies in mice with acute cellular rejection have demonstrated that MMP9 mediates the inflammatory response and triggers an immunological reaction that causes activation of pro-inflammatory cytokines and monocytes, and induces a series of changes in the extracellular matrix composition of the transplanted heart. With this influx, overexpression of TIMP1 is triggered, which has the role of protecting the affected myocardium through its anti-inflammatory potency. The occurrence of these dysfunctions is also due to the imbalance of MMP-TIMP, and the role of these inhibitors in attenuating possible myocardial damage caused by the proteolytic activity of these proteases is well known. However, this area of research is still active and new directions of approach are needed to study exactly how these inhibitors act on MPPs, including in human subjects [4,65,66].

Another study in mice published in 2005 also demonstrated that MMP2 and MMP9 play an important antagonistic role in rejection pathology, and that overexpression or underexpression of these proteases is associated with higher cardiac graft rejection. Thus, cardiac allografts from one group of mice with elevated MMP2 showed a more pronounced inflammatory and fibrotic response when compared to allografts harvested from MMP2 negative mice, the latter also reporting a significant increase in graft survival time. In contrast, MMP9 deficiency is associated with a lower graft survival rate, and overexpression of these proteases leads to lower cellular infiltration and fibrosis in grafts harvested from mice during the same period. Increased levels of MMP2 and MMP9 were experimentally measured in transplanted mice during cardiac graft rejection by RT-PCR. The occurrence of humoral rejection also correlates with the time interval after surgery. Hence, allografts transplanted in MMP2 negative mice were rejected later compared to allografts transplanted in MMP2 positive mice, and the opposite for MMP9 negative and positive mice, respectively. The cardiac tissue remodeling process represented by collagen deposition and mononuclear inflammatory infiltrate were in evidence in MMP2 positive graft rejection in the same study. Consequently, increased levels of MMP9 were associated with decreased levels of collagen, and consequently minimal mononuclear inflammatory infiltrate. MMP2 deficiency was associated with decreased T-cell alloreactivity, in contrast to MMP9 deficiency which showed the opposite, significantly increasing dendritic cell stimulation and T-cell responsiveness [9].

Regarding the contribution of these proteases in modulating cytokine activity, decreased cytokine production (especially IL 4, IL5, IL 6, -IL9, TNF alpha, and IF gamma) was observed in grafts from MMP2 negative mice, and increased production of the same cytokines in MMP9 negative grafts. Numerous studies on rejection have demonstrated an increased expression of MMP2 and MMP9, with these proteases acting specifically and influencing in particular the degree of acute humoral rejection [4,67,68].

A study in Japanese monkeys by Jun-ichi Suzuki et al. (2000) followed the expression of MMPs in acute and chronic cardiac rejection on days 1, 7, 22, 28, 40, 41, and 95 post-transplant. MMP2, MMP9, and TIMP1 antibodies were evaluated immunohistochemically. The native heart showed no expression of these antibodies. Instead, immunohistochemical expression was detected in allografts between days 7 and 95. Intense and diffuse MMP2 positivity was observed in mononuclear cells and cardiac fibers correlated in direct proportion to the degree of inflammation, and in inverse proportion to the degree of associated fibrosis. MMP9 also showed expression, but with limited distribution. In contrast, TIMP1 and TIMP2 had low expression. The expression of MMP2 and MMP9 was also determined by RT-PCR, with similar results [4].

MMP2 and MMP9 activity was also shown in small intramyocardial vessels affected by rejection [4]. These various animal studies have shown the contributions of MMPs in worsening the degree of cardiac rejection and their involvement in pathological processes associated with cellular rejection, but questions remain about the mechanism of action of these proteases in human subjects.

The contribution of MMPs in ischemia and reperfusion lesions (another cardiac post-transplant complication) especially MMP2 and MMP9, has also been mentioned. A study published by Po-Yin Chung et al. in rats showed a marked increase in MMP2 in the reperfusion process after ischemia, which resulted in cardiac mechanical dysfunction. The study of potential inhibitors of these MMPs could be an effective pharmacological strategy in the treatment of ischemia and reperfusion injury [69].

The involvement of MMP2, MMP9, and TIMP1 inhibitors in cardiac remodeling and their essential role in ischaemia and reperfusion injury was also highlighted in a study on the human heart published in 2005. The study was performed on 15 patients with unstable angina who were submitted for CABG and CPB, respectively, from whom myocardial tissue was sampled. The right atrial activity of reperfused myocardium was followed, showing a rapid increase in MMP2 and MMP9 activity accompanied by a rapid decrease in TIMP1, demonstrating global alteration of left ventricular function and persistence of ischemia. Increased serum MMP levels at 24 h postoperatively were also detected in the plasma of these patients [70]. Although MMP2 and MMP9 are considered the main MMPs with implications in the pathology of cardiac rejection, more detailed research of these families of proteases is necessary to establish more confident prognoses for these patients in the future.

Numerous studies have highlighted the role of these two proteases in the development of long-term heart failure. Increased MMP2 activity has been observed in the myocardium of hypertensive mice with heart failure. Increased MMP2 and MMP9 levels are known in patients with dilated cardiomyopathy [7,71].

The harmful effect of these proteases in degrading extracellular matrix components, in particular the activity of contractile proteins in the myocardium, is demonstrated in a study in rats, where MMP2 was observed to sit on the regulatory element of troponin I and to contribute to its degradation. Also, a study in rats reported the cleavage of sarcomeric myocardial proteins by MMP2 in ischemia and reperfusion injury, and the cardioprotective effect of carvedilol and nebivolol in reducing the activity of these proteases [72].

In addition to the involvement of these proteases in ischemia and reperfusion injury, their increased activity in the cardiac allograft rejection process is discussed. Thus, Aharinejad et al. published a study on 66 heart failure patients who underwent cardiac transplantation and from whom serial endomyocardial biopsies were taken at different times postoperatively (1, 2, 3, 4, 7, 12, 24, and 52 weeks), as well as serum samples monitoring MMP activity. On serum assessments, a maximum increase in MMP1, MMP8, MMP9, and TIMP1 concentrations was observed in the first 1–3 weeks, followed by a steady increase thereafter. Cellular rejection was also observed in about half of the patients in the study at a late post-transplant stage. The host immunological response may result in acute or chronic graft rejection potentiated by MMPs, with MMP1 responsible in this study [73].

The large majority of studies report a significant increase in MMP activity in the immediate post-transplant period, followed by a decrease, precisely due to immediate injury and the processes of ischemia and reperfusion in the transplanted heart [71,74,75,76,77].

In both animal and human models it has been shown that increased intramyocardial MMP levels are time-dependent and can thus lead to left ventricular dysfunction, with structural alteration of the extracellular matrix and initiation of ventricular remodeling processes with progression to end-stage cardiac disease [16,78,79,80,81]. The activity of these proteases is, however, modulated by TIMPs, making them a potential treatment modality for myocardial dysfunction, which needs to be further evaluated in the future.

## 8. Future Directions

The domain of cardiac transplantation is an intriguing topic in the literature with regards to the phenomenon of rejection, and the analysis of the impact of different types of MMPs on cardiac remodeling and the study of factors influencing the activity of these proteases is ever more important. Understanding these mechanisms allows the highlighting of potential changes in post-transplant myocardial structure induced by the activity of these MMPs, in order to improve post-transplant pharmacological therapy and quality of life of these patients.

Future research should focus on further investigating these incompletely elucidated mechanisms and monitoring these changes in transplanted human subjects, including large cohorts of patients and taking into consideration multiple factors for a more complete study design with a better chance of fully understanding this topic.

## 9. Conclusions

The increasing involvement of these proteases in a multitude of pathologies, perhaps by far the most well-known being cardiovascular pathology, has led to their activity being intensively studied in both animal models and human subjects.

MMPs are an important factor contributing to the alteration of multiple pathophysiological mechanisms, including angiogenesis, apoptosis, immune mechanisms, and tumor processes.

Special attention is focused on the presence of TIMPs, which are designed to control the activity of these proteases, which are extremely complex and still require detailed research.

## Figures and Tables

**Figure 1 medicina-59-01295-f001:**
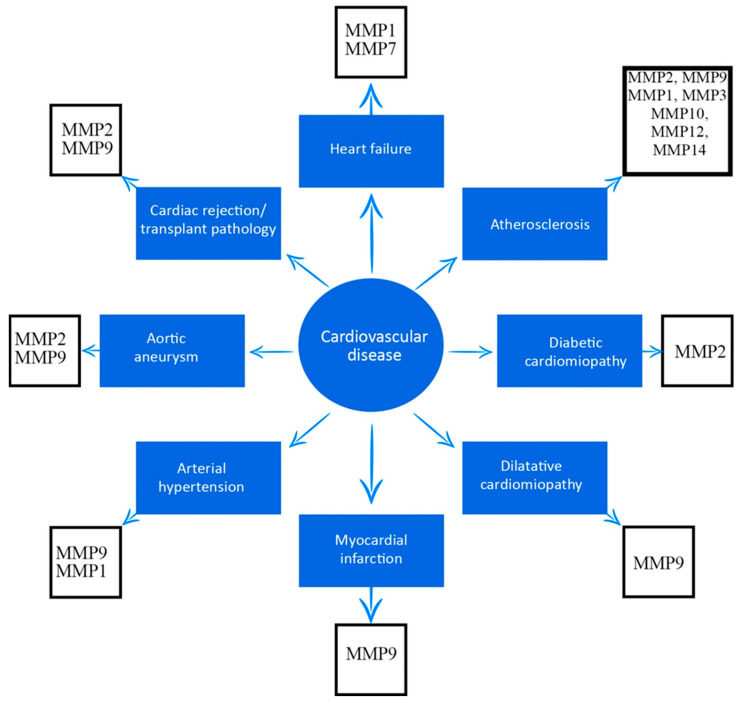
Association between matrix metalloproteinases and cardiovascular diseases.

**Table 1 medicina-59-01295-t001:** Main matrix metalloproteinases involved in cardiovascular pathology based on their function, location and structure.

Enzyme Name	MMP Classification (Number)	Common Description/Substrate	Cell	Cell Localization	Functional Role	Inhibition	References
Collagenase	MMP1, MMP8, MMP13, MMP18	Collagen II, III, VIII, X, gelatins, aggrecan, entactin	Myocytes, macrophages, fibroblasts, megakaryocytes, cardiomyocytes, neutrophils	Cytosol (MMP18) Nucleus (MMP1)	Myocardial fibrosis Inflammation	TIMP-1	[1] [11] [6] [18]
Gelatinase	MMP2	Gelatins Collagen I, IV, V, VII Basal membrane component	Cardiomyocytes Smooth muscle Platelet Megakaryocytes Fibroblasts	Sarcoplasmic reticulum Nucleus cytosol Mitochondria Sarcomere	Cardiomyocytes death Heart failure Increase angiogenesis, inhibition of myocardial fibrosis	TIMP 1-4	[18] [11]
	MMP9	Gelatins Collagen IV, V, XIV Basal membrane component	Leucocytes Megakaryocytes Cardiomyocytes	Cytosol Nucleus Sarcomere Mitochondria	Induce cardiac disfunction Increase migration rate of cardiac fibroblasts	TIMP1-3	[11] [20]
Stromelysines	MMP3, MMP10, MMP11	Collagen III, IV, V, IX Fibronectin Laminin	Megakaryocytes Cardiomyocytes	Cytosol nucleus	Cardiomyocyte death Increase heart failure Increase migration rate of cardiac fibroblasts	TIMP1-3	[6] [19]
Matrilysines	MMP7 MMP26	Fibronectin Collagen IV gelatins	Cardiomyocytes Neuronal cells	Cytosol	Increase migration rate of cardiac fibroblasts Inflammation	TIMP4 TIMP1	[11] [18]

## Data Availability

Not applicable.

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
