# Peer review of "Matrix Metalloproteinases and Heart Transplantation—A Pathophysiological and Clinical View"

_medicina, 2023, doi:10.3390/medicina59071295_

Round 1

Reviewer 1 Report

Dear authors,

A fascinating area of research in cardiovascular medicine is MMPs. Regarding the potential link between MMPs and heart transplant rejection, I congratulate you on this revision.

I would only suggest that you reduce the size of the article, focussing more on what is known today regarding MMPs in Heart Transplantion, and less extensively in the MMP general role, classification relationship with other cardiovascular disease.

Author Response

Estimeed reviewer,

Thank you very much for your highly appreciated recommendations.

According to this suggestion, the text was restructured and certain paragraphs were excluded.

Thank you very much for your feedback on our manuscript, we deeply appreciated and find very reasonable your observations.

The authors

Reviewer 2 Report

The manuscript entitled “Matrix Metalloproteinases and Heart Transplantation - A Pathophysiological and Clinical View” by Patrichi G et al is an interest review about the role of metalloproteinases in cardiovascular disease, mainly in post-transplant complications.

Major comments to the authors

1. The authors must carefully check the grammatical expression. Many sentences are not clear throughout the manuscript. 2. In MMPs introduction section the authors only mention activity regulation by TIMPs as the only mechanism. They should mention others. 3- line 88: authors indicate only intracellular MMPs as responsible of CVD, when extracellular MMPs are important actors. 4- Authors should remove line 90 to 92. 5- In line 115 authors mention that “The gelatinases MMP2 and MMP9 are known to play a role in angiogenesis and neurogenesis”. It is important to mention atherosclerosis process. 6-Line 225-229. The authors express that MMPs “destroy” vascular walls and atherosclerotic plaque. What is the meaning? MMPs, such as type 2 and 9, are involved both in development and vulnerability plaque. Please clarify this point (Cells. 2019 Feb 14;8(2):158. Doi: 10.3390/cells8020158) 7- The manuscript has a lack of connection in some points, for example, in page 8, hypertension condition suddenly appears when it has not been mentioned previously. Maybe, point 6 should has subheadings to better organize the reading. 8-Discussion. A first paragraph introducing the topic of revision is needed. Line 313: What is ISHLT abbreviature? Discussion section seems to be a continue of the revision. Authors should reconsider it.

Moreover, they mention the importance of MMPs measurement as biomarkers, however actually there are many limitations for use them for clinical diagnosis. Authors should describe this point.

As this reviewer mentioned in Comments to the authors, many sentences are not clear and some consecutive pharagraphs are not connected. 

Author Response

Answer to reviewer,

Esteemed reviewer,

Thank you for providing us the opportunity to submit a revised draft of our manuscript titled “Matrix Metalloproteinases and Heart Transplantation - A Pathophysiological and Clinical View”. We appreciate the time and effort that you have dedicated to providing your valuable feedback on our manuscript.

We have been able to incorporate changes to reflect most of the suggestions provided, we have highlighted in red the changes within the manuscript:

Comment 1: The authors must carefully check the grammatical expression. Many sentences are not clear

throughout the manuscript.

Response:  We sincerely appreciate the reviewer’s comments. According to your suggestion, the manuscript was proofread for proper English language, grammar, punctuation, spelling, and overall style by  Cambridge Proofreading & Editing LLC.

 We hope the revised manuscript will meet the requirements of academic publishing in Medicina.

Comment 2: In MMPs introduction section the authors only mention activity regulation by TIMPs as the only mechanism. They should mention others.

Response: Indeed, we strongly agree that the notions in the MMP introduction section on the regulation and activation of these proteases were incomplete, thus we mentioned in lines 90-94 the other pathways of MMP induction.

Comment 3: - line 88: authors indicate only intracellular MMPs as responsible of CVD, when extracellular MMPs are important actors.

Response: We agree with the reviewer’s opinion. We have modified the text in lines 88-89, as it is very important to mention the implications of these extracellularly localized proteases in cardiovascular pathology. In fact, the first decades of research on the role of these proteases in cardiovascular diseases have focused on this localization. Furthermore, chapter  5 has been restructured, where we mentioned the contribution of extracellular versus intracellular MMPs in cardiovascular pathology.

Comment 4: Authors should remove line 90 to 92.

Response: Thank you very much for your comment, lines 90-92 have been deleted from the manuscript.

Comment 5: - In line 115 authors mention that “The gelatinases MMP2 and MMP9 are known to play a role in angiogenesis and neurogenesis”. It is important to mention atherosclerosis process.

Response: Thank you for the suggestion. We strongly agree that it is important to mention the atherosclerotic process, as there are numerous studies that mention the involvement of metalloproteinases in this pathology, one of the most representative research on this topic being added in the bibliography (index citation no. 17, Berg et al). Thus, according to the reviewer suggestion, the text was modified in lines 110-114.

Comment 6: -Line 225-229. The authors express that MMPs “destroy” vascular walls and atherosclerotic plaque. What is the meaning? MMPs, such as type 2 and 9, are involved both in development and vulnerability plaque. Please clarify this point (Cells. 2019 Feb 14;8(2):158. Doi: 10.3390/cells8020158)

Response: Thank you very much for your comment. The specified lines were modified according to reviewer suggestion; the intention of the authors was to emphasize the role of MMP 2 and MMP 9 in the context of specific pathological processes, such as atherosclerosis and inflammatory vascular disease. Also, the suggested research was included in the bibliography. [index citation no. 14, Berg et al, and index citation 44, Provenzano M et al, lines 231-236 ]

Comment 7: The manuscript has a lack of connection in some points, for example, in page 8, hypertension condition suddenly appears when it has not been mentioned previously. Maybe, point 6 should has subheadings to better organize the reading.

Response: Thank you for your valuable opinion. Accordingly I have subdivided Chapter 6 as the reviewers suggested.

Comment 8: Discussion. A first paragraph introducing the topic of revision is needed.

Line 313: What is ISHLT abbreviature ? Discussion section seems to be a continue of the revision. Authors should reconsider it. Moreover, they mention the importance of MMPs measurement as biomarkers, however actually there are many limitations for use them for clinical diagnosis. Authors should describe this point.

Response: Thank you for the suggestion. The Discussion section was restructured, an introducing paragraph was added and ISHLT abbreviature was explained. Also, the phrase regarding the possible use of MMP serum levels was completed, to clarifiy this aspect (lines 351-359). 

Thank you very much for your feedback on our manuscript, we deeply appreciated and find very reasonable your observations.

The authors

Round 2

Reviewer 2 Report

The authors have answered all comments correctly.